# The Impact of Urinary Incontinence on Quality of Life: A Cross-Sectional Study in the Metropolitan City of Naples

**DOI:** 10.3390/geriatrics5040096

**Published:** 2020-11-20

**Authors:** Bruno Corrado, Benedetto Giardulli, Francesco Polito, Salvatore Aprea, Mariangela Lanzano, Concetta Dodaro

**Affiliations:** 1Department of Public Health, University Federico II of Naples, 80138 Naples, Italy; bruno.corrado@unina.it (B.C.); francescopolito94@gmail.com (F.P.); sa.aprea@gmail.com (S.A.); mariangelalanzano@gmail.com (M.L.); 2Department of Advanced Biomedical Sciences, University Federico II of Naples, 80138 Naples, Italy; cododaro@unina.it

**Keywords:** incontinence, quality of life, survey, questionnaire, patient-centered care, symptom perception, chronic illness

## Abstract

Urinary incontinence is a hygienic and psychosocial problem that often brings people to restrict their social life and to experience depression. The main aim of this study was to evaluate the impact of urinary incontinence on quality of life among residents of the Metropolitan City of Naples, Italy, using a newly designed multidimensional questionnaire. The secondary objective was to find which variables affect the quality of life and symptom severity in these patients. To do so, a sample composed of twenty-eight patients was recruited in a multicentre cross-sectional study. Most of the participants had a mild impairment (60%) concerning social life and self-perception, especially those whose education was above the primary level (*p* = 0.036) and those who followed a pelvic floor rehabilitation program (*p* = 0.002). Overflow urinary incontinence was associated with a greater deterioration in the aspirational and occupational domain (*p* = 0.044). Symptom severity was worse in those who had comorbidities (*p* = 0.038), who had a high body mass index (*p* = 0.008) or who used diuretics (*p* = 0.007). In conclusion, our results suggest that there is a significant impairment of quality of life in patients who have only primary education and who follow a pelvic floor rehabilitation program.

## 1. Introduction

Urinary incontinence (UI) is a hygienic and psychosocial problem characterised by the involuntary loss of urine, which is commonly associated with sneezing, coughing or a sense of urgency and which can cause stress and fatigue in patients [1]. More than 200 million people worldwide experience UI; this problem is more common in women than in men [2], and prevalence increases with age [3]. There are different types of UI, each with distinct symptoms and causes: stress, urgency, mixed and overflow urinary incontinence [4,5]. Treatment of UI depends on the type of symptoms and underlying pathology [6], and often it requires a multidisciplinary team in order to take care of these patients [7].

UI influences many aspects of patients’ lives such as family and work relationships and sexual function, and therefore, it may decrease patients’ quality of life (QOL). UI is often hard to accept for people who suffer from it, due to its negative impact on their privacy and sexuality. Indeed, many people are embarrassed to disclose this problem and do not consult doctors for management and treatment [8]. Discomfort caused by feeling “wet”, “dirty” or “smelly” leads many persons to restrict their social and physical activities such as exercise, shopping, dancing, going to church and visiting friends. In addition, UI patients often experience low self-esteem, low physical attractiveness, body negativity [9], inadequacy, stigmatisation and a reduction of sexual desire, which can lead to loneliness, self-isolation and depression [10]. In the most severe cases, even activities of daily life are compromised since patients must continually interrupt what they are doing to use the bathroom [11,12]. In addition to all the negative medical and psychosocial aspects, the high economic burden of UI is a factor to be considered: the money spent for pads is an important cost item [13].

Many factors affect QOL in patients with UI: condition severity, age, self-perception, sex, comorbidities, etc. If other medical conditions are added to UI, e.g., obesity, QOL will be even lower because of worsening of the incontinence itself and because of the social stigmas of obesity [9] and of the disease itself [14].

Studies from the scientific literature have shown that several coping strategies are associated with better QOL in patients suffering from chronic diseases such as UI: active coping, acceptance of the disease, being optimistic, expression of feelings and positive religious coping [15]. In the same way, other studies have revealed that behavioural disengagement, self-blame, high distraction and negative religious coping such as a pessimistic view of the world and spiritual discomfort are related to worse QOL.

Individuals in the patient’s social network (e.g., proxy, caregiver, family) may sometimes be needed to participate in an intervention of health promotion. The patient’s partner has an important role because UI affects the couple’s QOL too—sexual satisfaction—since UI may inhibit sexual activities through association with embarrassment and shame [16,17]. It is important to help the couple to live serenely with the symptoms.

The primary objective of this study was to investigate the impact of UI on QOL among residents of the Metropolitan City of Naples, Italy. The secondary objective was to determine which variables affect the QOL and symptom severity in these patients.

## 2. Materials and Methods

### 2.1. Design

We conducted a multicentre cross-sectional study on a small sample—brief report—affected by UI, which is a representative subset of a given socio-cultural context, in order to obtain a snapshot of QOL perception and symptom severity. The study methods were compliant with the strengthening the reporting of observational studies in epidemiology (STROBE) checklist (see Appendix A). All patients’ data were recorded anonymously. Finally, we analysed the data obtained and offer some hypothesis of correlation, which could be investigated in further studies.

### 2.2. Material

To assess QOL and severity of UI symptoms in patients we built a multidimensional specific questionnaire, the Quality of Life Questionnaire: Urinary Incontinence (QOL-UI; see Appendix A). We chose not to use a validated questionnaire already existing in the literature [18] because we wanted a survey that could provide a wide assessment of patients’ life perceptions specifically for UI. For example, though the International Consultation on Incontinence Questionnaire-Urinary Incontinence Short Form (ICIQ-UI SF) is one used most commonly worldwide for the evaluation of the frequency, severity and impact on QOL of UI in men and women [19], we did not use it because, due to unidimensionality, it did not allow a holistic assessment of the patient considering his physical, emotional, social, economic and spiritual needs.

### 2.3. Questionnaire Structure

The questionnaire included data about the patient’s general information (gender, age, occupation and education), anthropometric data (height, weight and body mass index [BMI]) and data relating to the patient’s general health status (comorbidities, hospitalisation events, medications taken and motor deficits). The questionnaire also contained questions that assessed UI severity and other factors (e.g., number of childbirths in women). The second part of the questionnaire was directed towards the clinicians, with the intention to indicate the type of UI based on the information reported in the previous section and quantify the severity of the condition. This section was not intended to be filled in by patients. The assessment was made by associating the answers given to nine items that have a numerical score of varying severity:Walking autonomy: this autonomy was assessed in order to identify those patients who were physically unable to independently reach the nearest toilet. The item assessed the walking ability and not the quality of gait; therefore, if the patients used orthopaedic or walking aids that supported them in walking, they were considered nonautonomous. In the absence of walking autonomy, a score of 5 was assigned, otherwise, 0.Frequency of urination per day: the option “less than 6 times” corresponded to 0, “from 6 to 10 times” corresponded to 1, “from 10 to 15 times” corresponded to 2 and “more than 15 times a day” corresponded to 3.Losses during the day: “episodes of UI during the day” or “episodes of UI during the night” options corresponded to 1, while the “both day and night” option corresponded to a severity score of 2.Frequency of the episodes of UI: the “rarely” option corresponded to 1, “once or more times a week” corresponded to 2, “once a day” corresponded to 3 and “more than once a day” corresponded to 4.Amount of urine leakage: the question posed four images of a medium-sized sanitary napkin with a variable wet surface. The numerical scores varied respectively from 1 for the least absorption of liquid to 4 for complete saturation.Sphincter control during urination: if the patient was able to stop the flow of urine when urination has started, a score of 0 was be assigned, otherwise, 1.Circumstances in which the patient had involuntary leakage of urine: for each circumstance that led the patient to lose urine, 1 point of severity of UI was assigned.Absorbent aids used by the patient: a score of 0 was assigned if the patient did not use aids at all, 1 point if he or she used a draw sheet, 2 points if he or she used a diaper or absorbent pad or both and 3 points if he or she used a catheter.Number of changes of aids made per day (every 24 h): the option “1 change per day” corresponded to 1, “2 changes per day” corresponded to 2, “3 changes per day” corresponded to 3 and “more than 3 changes per day” corresponded to 4.

The total score for UI severity ranged between 5 and 32. This section needed to be filled in once the patient had completed the other sections of the questionnaire.

The third and final part of the questionnaire corresponded to the actual QOL questionnaire for UI. The questionnaire had 35 items that the patient answered using an attitude scale, or Likert scale, with four answer options, from “Not at all” to “Very much”. The Likert scale was subsequently converted by the clinicians into a numerical scale: the answer option “Not at all” corresponded to 1 point and “Very much” to 4 points. The optimal QOL score was therefore 35 points out of the maximum possible 140. The questionnaire was constructed to investigate numerous domains of QOL specifically concerning UI (Figure 1). We summarized these into three macrodomains. “Being” encompassed organic, physiologic and spiritual aspects and had 17 items (range 17–68). “Belonging” encompassed physical, social and public aspects and had nine items (range 9–36). “Becoming” encompassed practical, recreational and ambition-related aspects and had nine items (range 9–36). Items were arranged so as not to make the domains obvious. Clinicians calculated the domain scores using the rating system shown in Figure 1.

Sample scores were classified into three severity levels: slight (5–10), mild (11–20) and severe (21–32). The QOL score and its domains were classified similarly: slight impairment (30–65), mild impairment (66–100) and severe impairment (101–140); for “Being”, the corresponding ranges are 17–34, 35–51 and 52–68, and for both “Belonging” and the “Becoming”, they are 9–18, 19–27 and 28–36.

The estimated time of compilation was 5 to 10 min. The instrument was intended for patients with a history of UI for at least 1 year and age of at least 18 years.

No urodynamic exam was required to verify the type of UI.

### 2.4. Data Collection

Twenty-eight patients with a confirmed UI diagnosis were recruited and assessed with the QOL-UI, all at least 18 years old. Three participants were excluded because they did not fill in the questionnaire correctly. No exclusion due to cognitive impairments was made. The sample consisted of 25 patients, 10 male and 15 female. The recruitment took place at Ufficio Prescrizione Ausili Assorbenti in District 44 of the Azienda Sanitaria Locale Napoli 2 Nord, at the Reparto di Urologia and at the Ambulatorio di Urodinamica at Azienda Ospedaliera di Rilievo Nazionale A. Cardarelli.

Inclusion criteria: diagnosis of UI, age superior to 18 years, time from the onset of UI more than 1 year, under treatment by the Italian National Health System.

Exclusion criteria: questionnaires not entirely filled, cognitive functions preserved enough to understand the meaning of the study and to fill in the questionnaire.

### 2.5. Analysis

The sample has been stratified based on different category variables, in order to make a comparison between the medians of two or more independent groups. To do so, nonparametric tests, like Kruskal-Wallis and Mann-Whitney with a *p*-value threshold of 0.05, were used. The analysis studied ordinal and continuous variables, characterised, respectively, by numbers and percentages and by means and standard deviations. To study comorbidity, the absolute frequency and the relative frequency in percentage were calculated. All data were entered and analysed using Microsoft Excel version 2013. To verify the correlations between continuous variables, the Pearson correlation coefficient (r) and determination coefficient (r^2^) in the linear regression graph were calculated. Since the study presented is a brief report, no sample size calculator has been used.

### 2.6. Sample Characteristics

The sample had a majority of women (60%), and about 84% of the participants were at least 50 years old. Of the participants, 28% had received an education above the primary level, 16% declared that they were employed and 80% had a BMI greater than the regular cut-off point (25.0). The type of incontinence most represented was the mixed one (95%). Other demographic and clinical characteristics are reported in Table 1. The main causes of UI (Figure 2) were, for women, urogenital prolapse (44%) and, for men, prostatic hypertrophy associated with overfull bladder (16%); however, no main cause was found for 16%. Comorbidity was common in the sample, and the most frequent condition was hypertension (56%), followed by diabetes (40%). Most of the participants had been suffering from UI for more than 5 years (40%), whereas 28% had been suffering for just over 1 year. The 44% of participants reported changing absorbent aids more than three times per day, while 24% of the sample did not use absorbent aids. Further clinical characteristics are in Table 2.

### 2.7. Ethics

The study was conducted anonymously in compliance with the 2005 version of the Helsinki Statement. All patients gave informed consent. Ethical approval was not required but the study was carried out under the supervision of the local ethics committee anyway.

## 3. Results

The analysed sample, for the most, presented moderate severity of the symptoms (mean 17.6; DS 4.36; Table 3) and a QOL total score indicating moderate compromise (mean 91.24; DS 20.11; Table 3), but the influence of severity on QOL was not been significantly appreciated (r = 0.30). Severity was affected by age (r = 0.59, *p* = 0.001) and BMI (r = 0.58, *p* = 0.002), according to the Pearson correlation analysis (Table 3), and by comorbidities (U = 24, *p* = 0.038), absorbent aids use (U = 22.5, *p* = 0.030), diuretic use (U = 24, *p* = 0.007) and BMI again (K = 9.47, *p* = 0.008), according to nonparametric tests (Table 4).

QOL total score, according to nonparametric tests (Table 4), was affected by the educational level (U = 28, *p* = 0.036) and pelvic floor rehabilitation (U = 8, *p* = 0.002). No positive significant correlation was found either between QOL total score and age in both sexes (r = 0.13) or between QOL total score and age among women (r = 0.02). Instead, a moderate positive correlation was found between QOL total score and age among men (r = 0.53; Figure 3).

Analysing the QOL domains (Table 5), we found that patients presented, for the most part, less impairment in the “Belonging” domain (slight in 52% of the sample), while the “Being” and “Becoming” domains were more compromised (mild impairment in 56% and 52% of participants, respectively). Using nonparametric tests on QOL domains (Table 6), we also found that the “Being” domain was more compromised in those who had a low educational level (U = 24, *p* = 0.026), the “Becoming” domain was more compromised in men (U = 39, *p* = 0.048) and in overflow UI type (K = 6.20, *p* = 0.044) all domains were more compromised in those who followed a pelvic floor rehabilitation program (U = 11.5, *p* = 0.009; U = 8.5, *p* = 0.002; U = 9, *p* = 0.002; “Being”, “Belonging” and “Becoming”, respectively).

## 4. Discussion

This study investigated QOL, its domains and the severity of UI symptoms in relation to clinical or demographic variables in a sample of residents of the Metropolitan City of Naples. UI is a cause of embarrassment, especially for women, because it can interfere with self-image and consequently also self-esteem and sense of attractiveness [20]. The “Being” domain, which includes safety and self-concept, was moderately compromised in 73.3% of women and severely compromised in 20% of cases. The same domain deterioration was observed in men. Low self-esteem and a negative body image were also associated with a low level of education [21]. Indeed, in the study there was considerable impairment of the “Being” domain in those whose highest educational level completed was primary school.

Another significantly greater impairment was found in males in relation to the “Becoming” domain, which included work, self-determination and interests. This could be attributed to the fact that most of the women were unemployed or housewives, and therefore, UI did not have a negative impact on their job. In addition, the study showed that for men with the same symptom severity as women there was greater impairment of QOL.

BMI is strongly correlated with the severity of UI symptoms [22] and consequently with QOL, because the increase of intra-abdominal pressure causes weakness of pelvic floor muscles and fascia. This study has indeed confirmed that a higher BMI is associated with more severe UI symptoms. On the other hand, no greater impairment of QOL was observed in subjects with higher BMI, although the negative effects of stigma and social exclusion upon body image and self-esteem of subjects with obesity are recognized.

Comorbidities compromise homeostasis, so they may potentially aggravate UI symptoms. In the study, 76% of the sample presented clinical comorbidities that were associated with significantly worse scores for severity and the “Being” domain. Of these, 64% suffered from high blood pressure treated with diuretics, which was seen to be associated with higher UI severity. Therefore, this suggests that diuretic drugs cause more severe UI symptoms and consequent greater impairment of QOL. This relationship can be explained by the increase of the volume of urine produced and therefore by the increase in both the number of times the patient goes to the bathroom and the amount of involuntary urine leakage. These events would lead to the pathogenesis of overactive bladder, especially even in the presence of a high BMI [23].

Absorbent aids play a fundamental role in changing the body image and self-esteem of subjects with UI [8,24,25]. Participants reported several times that they had to change their clothing when wearing an absorbent aid. Therefore, there is a compromise in the “Becoming” domain because self-determination could be lacking in these subjects. In some cases, even the “Belonging” domain is compromised, especially when subjects sacrifice their social life for fear that absorbent aids may give off bad smells. Instead, the participants who claimed to change aids more than three times a day had a more compromised domain of “Being” [26]. However, after being educated through a self-care program, some patients may positively accept the use of an aid like a long-term catheter because it could help them to retake control of their lives [27].

In the sample studied, 44% of the participants said they used more than three absorbent aids per day. This is interesting as it could give us a rough estimate of the direct costs of the National Health System (NHS) and also partly the indirect costs, as question number 16 investigates the financial impact that incontinence has had on subjects. The estimate of indirect costs remains clearly subjective, so it would be necessary to investigate in a further study the economic impact of UI, both on the NHS and on the patient. The disproportionate use of absorbent aids per day could be linked to improper use, related also to an inappropriate prescription process [28], to the poor compliance of the product or to the lack of correlation between pad use and severity of the incontinence [29]. If we add to this the unnecessary surgical procedures that many patients experience, which could be prevented by urodynamic investigation, the direct costs reimbursed by the Italian National Health System are even higher [30].

Sexual satisfaction and intimate relationships did not show significant impairment in this study, although several previous papers have stated how much UI affects the sexual sphere [31]. When completing the questionnaire, many participants declared that they had no interest in sex. Probably embarrassment played a decisive role in giving such a response, above all among older people. This figure remains to be further explored.

Mixed UI, in accordance with the literature [32], resulted in the highest severity scores, followed immediately by overflow incontinence, and finally by urgency incontinence. However, the data collected in this study are not statistically significant. Nonetheless, this is interesting from a clinical perspective. The clinical and epidemiological research on mixed incontinence has been largely ignored. Many intervention or treatment studies exclude subjects with mixed UI from their samples. In clinical reality, in fact, there is a lot of difficulty in prescribing treatment options [33]. In terms of compromising QOL, however, overflow was the type with the greatest impact, in particular in the “Becoming” domain. Further studies should be performed on larger samples in order to better understand the impact of various types of incontinence on QOL and the relationships between severity and type of incontinence.

The rehabilitation of the pelvic floor, carried out by physiotherapists, has been considered for years a valid tool able to improve UI symptoms. It reduces urine losses in a period that ranges from 3 to 5 months from the start of the rehabilitation program [34]. Each patient should follow a customised rehabilitation program that includes standard physiotherapy interventions that aim to reduce pain, improve the strength and coordination of the pelvic floor muscles and stabilise the lumbar and abdominal muscles [35]. Home programs are also important because they can improve subjective symptoms, while outpatient programs improve objective symptoms [36]. If the exercise program is combined with direct electric stimulation on pelvic nerves, a synergic effect can also be observed [37]. To obtain results, however, it is essential that there are no spinal cord injuries. In the study presented, following a pelvic floor rehabilitation program had a significantly greater impact on QOL in all domains. The data in itself cannot be easily interpreted because the duration and intensity of the rehabilitation program that patients followed is unknown. Moreover, it is not known if the data had shown objective or subjective changes over time. It would be necessary to examine the data in a dedicated study. However, at a first interpretation, the relation could be explained by the degree of condition awareness developed among patients who do rehabilitation. Another reason may be that rehabilitation is an elite therapy reserved for a few selected patients.

QOL is closely linked to age [38] because it depends on the point of our life we are at. In fact, the study revealed a slight correlation between QOL and age, especially in men, but also another slight correlation between QOL and severity of symptoms. During the assessment, the elderly often said they did not leave their home and that they felt like a burden for their family. They also avoided, more often than not, seeking help or supporting [39]. This last point is very important because good and effective communication is fundamental in today’s clinical care. Therefore, in urology and urogynecology, counseling, which is based on patient-centered communication, has become a gold standard for excellence. It, indeed, enables a personalized therapy program to be reached thanks to a shared understanding with patients [40]. Finally, a statistical significance was identified as regards the correlation between the time of onset of the symptoms and the severity. In the study, those who had suffered from UI for more than 5 years had greater severity of symptoms, and with it also a greater impairment of QOL [9]. This could be explained by a possible evolution of the clinical picture over time.

The study has several limitations. First of all, a newly designed, non-validated questionnaire was used; its validity and reliability would deserve further investigation. Second, a favorable sampling, that includes only subjects who voluntarily chose to participate in the study, was analysed, and therefore, there is selection bias. Lastly, the study was carried out on a small sample (<30); therefore, it would be interesting to extend the sample size to observe the generalizability of the results.

## 5. Conclusions

The majority of enrolled patients had moderate severity of symptoms and moderate impairment of QOL. In addition, a significant impairment of QOL, especially in the “Being” domain, was observed in both men and women who had received only primary education. Those who had followed a pelvic floor rehabilitation program had an impairment on all three components of QOL: “Being”, “Belonging” and “Becoming”. However, in men, the “Becoming” domain had a greater impairment compared to women. Furthermore, among the types of UI, overflow incontinence had a greater impact on the domain of “Becoming”. On the other hand, severity was seen to be more compromised in those who had a high BMI, had comorbidities, took diuretics or used more than three absorbent aids per day. Further studies are needed to better frame the scope of the problem and to investigate phenomena in a more targeted perspective.

UI can indirectly affect the physical, psychological and social aspects of both men and women because it may significantly interfere with patients’ family and social life. Symptoms should be carefully evaluated from a holistic and multiperspective point of view. Any therapeutic program should be personalised to match the patient’s individual needs. To this end, it is essential that an interdisciplinary team take care of patients’ health and related problems. Physicians, nurses, physiotherapists and, above all, psychologists have a central role in identifying health needs, and their collaboration is fundamental to the taking care of patients suffering from UI. In conclusion, holistic patient-centred care should be always preferred. This requires an approach oriented to emotions, to interests and, in general, to the whole person.

## Figures and Tables

**Figure 1 geriatrics-05-00096-f001:**
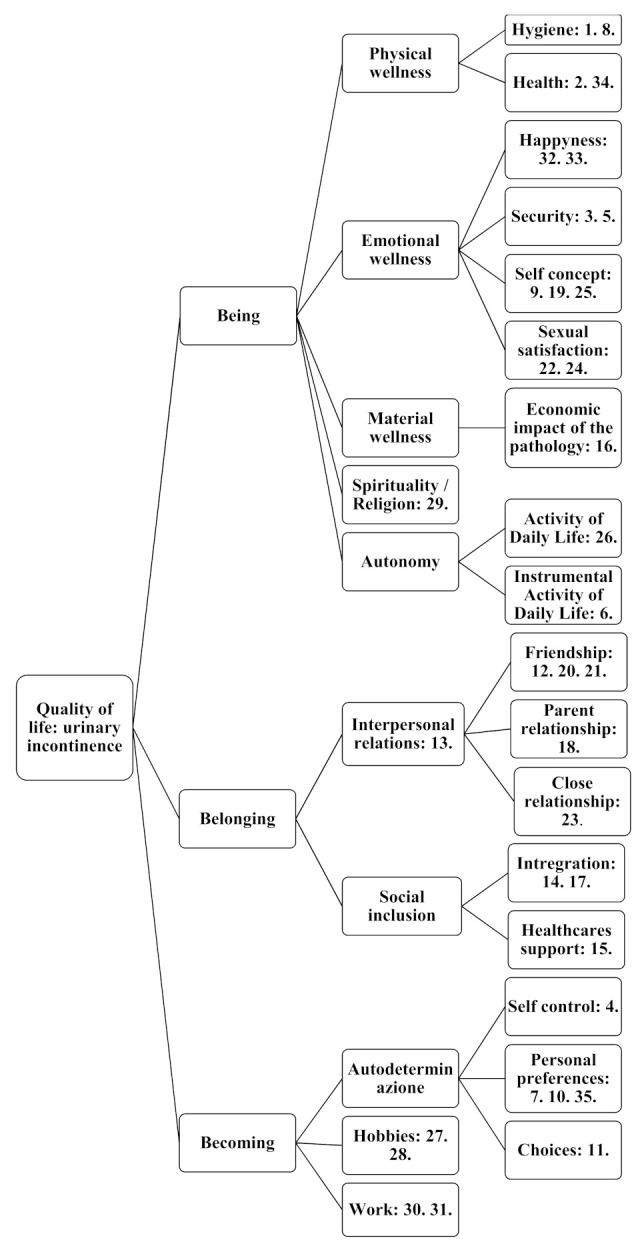
This figure shows the quality of life domains analysed using the questionnaire.

**Figure 2 geriatrics-05-00096-f002:**
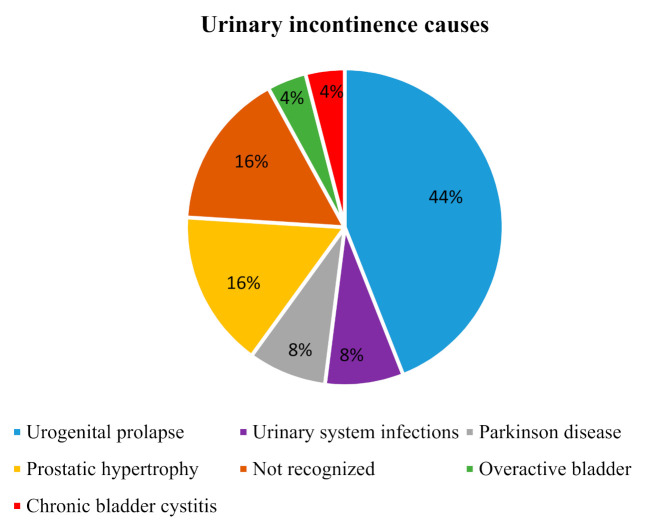
This figure shows the most represented urinary incontinence causes in the sample.

**Figure 3 geriatrics-05-00096-f003:**
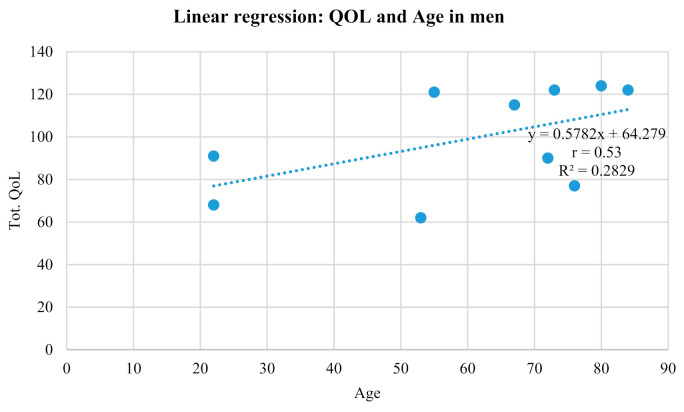
This figure shows the graphic of a linear regression between QOL scores and age in men.

**Table 1 geriatrics-05-00096-t001:** This table shows the demographic and clinical characteristics of the sample.

Demographic and Clinical Characteristics	N (%)
Age (years) (Mean, SD)	65.32 (DS 17.89)
Sex (female)	15 (60)
Women who gave birth (nulliparous)	13 (13.3)
Body Mass Index (Mean, SD)	30.55 (DS 5.97)
Workers (unemployed)	4 (84)
Education:	
Primary	18 (72)
Secondary	3 (12)
High	3 (12)
Higher	1 (4)
Physical activity (yes)	1 (4)
Medical comorbidity (present)	19 (76)
Deambulation autonomy (absent)	3 (12)
Incontinence type:	
Stress	1 (4)
Urgency	5 (20)
Mixed	14 (56)
Overflow	5 (20)
Functional	0

SD = standard deviation; N = number; % = percentage.

**Table 2 geriatrics-05-00096-t002:** This table shows the clinical characteristics of sample.

Further Clinical Characteristics	N	%
For how long has he/she been suffering from UI symptoms?		
Few weeks	0	0
Less than a year	0	0
Just over 1 year	7	28
Between 2 and 5 years	8	32
More than five years	10	40
Daily changes (24h):		
1	2	8
2	5	20
3	7	28
More than 3	11	44
How often he/she uses bathroom per day (24 h):		
Less than 6	2	8
From 6 to 10 times	12	48
From 10 to 15 times	5	20
More than 15 times	6	24
UI aids used:		
None	6	24
Traverse	1	4
Absorbent	14	56
Diaper	8	32
Catheter	1	4
Adverse drugs taken by patients with urinary incontinence:		
Diuretics	16	64
Laxatives	4	16
Sedatives	6	24
Muscle relaxants	1	4
None	6	24

N = number; % = percentage.

**Table 3 geriatrics-05-00096-t003:** This table shows the severity and quality of life (QOL) scores distributions with relative domains and mean correlations (N = 25).

Scales and Domains	Mean ± SD	Mi	Ma	Severity		Tot. QOL		The Being		The Belonging		The Becoming	
				r	*p*	r	*p*	r	*p*	r	*p*	r	*p*
Severity	17.6 ± 4.36	7	23	-		0.25	0.40	0.17	0.42	0.14	0.51	0.02	0.93
Age	65.32 ± 17.89	22	87	0.59	0.001 *	0.13	0.52	0.17	0.41	0.04	0.84	0.16	0.45
BMI	30.55 ± 5.92	20	40.2	0.58	0.002 *	0.08	0.69	0.07	0.74	0.17	0.41	0.17	0.40
Tot QoL	91.24 ± 20.11	58	124	0.25	0.40	-		-		-		-	
Being	47.52 ± 8.61	33	61	0.17	0.42	-		-		-		-	
Belonging	20.52 ± 7.71	10	34	0.14	0.51	-		-		-		-	
Becoming	23.2 ± 5.53	14	33	0.02	0.93	-		-		-		-	

Mi = minimum; Ma = maximum; r = correlation coefficient; * *p* < 0.05.

**Table 4 geriatrics-05-00096-t004:** This table shows the mean score comparison between severity and total QOL, based on socio-demographic and clinic characteristics.

Variables	N	%	Severity		Tot. QOL	
**Sex**						
Male	10	40	15.8 ± 5.41	U = 50	99.2 ± 24.45	U = 50.5
Female	15	60	18.8 ± 3.14	*p* = 0.174	85.93 ± 15.29	*p* = 0.183
**Age**						
<70	12	48	16 ± 5.20	U = 57	90.33 ± 19.12	U = 72.5
>70	13	52	19.08 ± 2.87	*p* = 0.262	92.07 ± 21.74	*p* = 0.787
**Education**						
Primary	18	72	18.5 ± 3.84	U = 38	96.5 ± 19.28	U = 28
>Primary	7	28	15.28 ± 5.06	*p* = 0.138	77.14 ± 16.4	*p* = 0.036 *
**Body Mass Index (BMI)**						
Regular <25	5	20	14 ± 5.29	K = 9.47	83 ± 20.61	K = 2.19
Overweight	7	28	15.43 ± 4.24	*p* = 0.008 *	102.28 ± 23.58	*p* = 0.317
Obese >30	13	52	20.15 ± 2.11		88.46 ± 16.78	
**Comorbidity**						
Present	19	76	18.89 ± 3.09	U = 24	93.73 ± 19.73	U = 39.5
Absent	6	24	13.5 ± 5.5	*p* = 0.038 *	83.33 ± 21.01	*p* = 0.280
**Childbirths number**						
1–3	5	20	20.2 ± 1.30	U = 45.5	90.2 ± 13.46	U = 9
+3	7	28	19.14 ± 3.85	*p* = 0.810	78.28 ± 12.34	*p* = 0.193
**Pelvic floor rehabilitation**						
Yes	6	24	16.33 ± 5	U = 41.5	114.67 ± 12.29	U = 8
No	19	76	18 ± 4.20	*p* = 0.954	83.84 ± 16.02	*p* = 0.002 *
**Types of incontinence**						
Urgency	5	20	14.6 ± 6.10	K = 4.53	94.6 ± 15.88	K = 5.35
Mixed	14	56	19.21 ± 3.26	*p* = 0.103	84.78 ± 16.50	*p* = 0.068
Overflow	5	20	17 ± 4		110.6 ± 22.95	
**For how long has he/she been suffering?**						
>1 year	7	28	18.14 ± 5.05	K = 5.76	88.14 ± 10.97	K = 0.28
>2 <5 years	8	32	15.37 ± 3.29	*p* = 0.055	90 ± 28.33	*p* = 0.865
>5 years	10	40	19 ± 4.29		94.4 ± 18.83	
**Daily changes (24 h)**						
2	5	20	15.6 ± 3.36	K = 5.64	82.6 ± 16.77	K = 0.91
3	7	28	18.43 ± 2.99	*p* = 0.059	91.28 ± 25.08	*p* = 0.634
More than 3	11	44	19.73 ± 3.10		92.45 ± 18.8	
**How often he/she uses bathroom per day (24 h)**						
6–10 times	12	48	17.66 ± 3.20	K = 1.02	89.66 ± 21.96	K = 0.31
10–15 times	5	20	19.6 ± 2.96	*p* = 0.600	97.4 ± 23.98	*p* = 0.853
>15 times	6	24	16.5 ± 6.80		92 ± 16.78	
**Use of aids**						
No	6	24	13.16 ± 5.6	U = 22.5	82.33 ± 21.31	U = 33.5
Yes	19	76	19 ± 2.83	*p* = 0.030 *	94.05 ± 19.45	*p* = 0.144
**Use of diuretics**						
No	9	36	14.33 ± 4.62	U = 24	85.44 ± 18.17	U = 54.5
Yes	16	64	19.44 ± 3.01	*p* = 007 *	94.5 ± 20.97	*p* = 0.337
**Use of sedatives**						
No	19	76	16.89 ± 4.71	U = 42.5	91.26 ± 18.61	U = 56
Yes	6	24	19.83 ± 1.83	*p* = 0.373	91.17 ± 26.35	*p* = 0.976

U = Mann‒Whitney U test; K = Kruskal‒Wallis test; * *p* < 0.05.

**Table 5 geriatrics-05-00096-t005:** This table shows the sample numbers and percentages divided into classes.

Classes	Severity			Tot. QOL			The Being			The Belonging			The Becoming		
	Range	N	%	Range	N	%	Range	N	%	Range	N	%	Range	N	%
**Total sample N = 25% = calculated on total sample**															
Slight	5–10	2	8	30–65	2	8	17–34	3	12	9–18	13	52	9–18	6	24
Mild	11–20	15	60	66–100	15	60	35–51	14	56	19–27	6	24	19–27	13	52
Severe	21–32	8	32	101–140	8	32	52–68	8	32	28–36	6	24	28–36	6	24
**Men N = 10% = calculated on male sample**															
Slight	21–32	2	20	30–65	1	10	17–34	2	20	9–18	3	30	9–18	1	10
Mild	11–20	6	60	66–100	4	40	35–51	3	30	19–27	2	20	19–27	4	40
Severe	21–32	2	20	101–140	5	50	52–68	5	50	28–36	5	50	28–36	5	50
**Women N = 15% = calculated on female sample**															
Slight	5–10	0	0	30–65	1	6.6	17–34	1	6.6	9–18	10	66.6	9–18	5	33.3
Mild	11–20	9	60	66–100	11	73.3	35–51	11	73.3	19–27	4	26.6	19–27	9	60
Severe	21–32	6	40	101–140	3	20	52–68	3	20	28–36	1	6.6	28–36	1	6.6

N = number; % = percentage.

**Table 6 geriatrics-05-00096-t006:** This table shows the mean score comparison between QOL domains, based on socio-demographic and clinic characteristics.

Variables	N	%	Being		Belonging		Becoming	
**Sex**								
Male	10	40	48.8 ± 10.99	U = 62	24.4 ± 8.14	U = 42.5	26 ± 6.05	U = 39
Female	15	60	46.6 ± 6.88	*p* = 0.490	17.93 ± 6.44	*p* = 0.075	21.33 ± 4.42	*p* = 0.048 *
**Age**								
<70	12	48	48.53 ± 8.74	U = 69	19.66 ± 7.14	U = 71.5	22.08 ± 4.83	U = 60.5
>70	13	52	46.54 ± 8.71	*p* = 0.462	21.31 ± 8.42	*p* = 0.326	24.23 ± 6.11	*p* = 0.924
**Education**								
Primary	18	72	50.22 ± 7.60	U = 24 *p* = 0.026 *	21.88 ± 7.96	U = 39.5 *p* = 0.164	24.38 ± 5.36	U = 36 *p* = 0.109
>Primary	7	28	40.57 ± 7.37		17 ± 6.19		20.14 ± 5.08	
**Body mass index (BMI)**								
Regular <25	5	20	42.8 ± 8.10	K = 2.87	19.2 ± 9.15	K = 1.70	21 ± 6.12	K = 2.79
Overweight	7	28	51.86 ± 8.99	*p* = 0.238	24.14 ± 8.78	*p* = 0.426	26.28 ± 5.99	*p* = 0.248
Obese >30	13	52	47 ± 8.08		19.08 ± 6.47		22.38 ± 4.73	
**Comorbidity**								
Present	19	76	49.10 ± 8.03	U = 33	20.79 ± 8.04	U = 56	23.84 ± 5.65	U = 42
Absent	6	24	42.5 ± 9.14	*p* = 0.133	19.67 ± 7.2	*p* = 0.976	21.17 ± 5.04	*p* = 0.357
**Childbirths numbers**								
1–3	5	20	49.4 ± 5.77	U = 11.5	19.4 ± 5.98	U = 7.5	21.4 ± 3.43	U = 11
+3	7	28	44.23 ± 8.28	*p* = 0.373	14.43 ± 2.64	*p* = 0.123	19.43 ± 3.41	*p* = 0.332
**Pelvic floor rehabilitation**								
Yes	6	24	55.17 ± 5.74	U = 16	28.83 ± 4.21	U = 8.5	29.67 ± 3.98	U = 9
No	19	76	45.10 ± 7.99	*p* = 0.009*	17.58 ± 6.02	*p* = 0.002*	21.16 ± 4.25	*p* = 0.002 *
**Types of incontinence**								
Urgency	5	20	46 ± 7.81	K = 4.12	23.8 ± 5.89	K = 5.84	24.8 ± 3.35	K = 6.20
Mixed	14	56	46.43 ± 7.53	*p* = 0.127	17.36 ± 6.7	*p* = 0.052	21 ± 4.77	*p* = 0.044 *
Overflow	5	20	54.8 ± 8.95		27.2 ± 8.04		28.6 ± 6.11	
**For how long has he/she been suffering?**								
>1 year	7	28	46.14 ± 6.01	K = 0.91	19.71 ± 5.44	K = 0.04	22.28 ± 3.35	K = 0.32
>2 <5 years	8	32	46 ± 12.22	*p* = 0.634	20.75 ± 8.73	*p* = 0.975	23.25 ± 7.57	*p* = 0.850
>5 years	10	40	49.7 ± 6.96		20.9 ± 8.89		23.8 ± 5.33	
**Daily changes (24 h)**								
2	5	20	42.6 ± 8.02	K = 2.22	18.2 ± 4.32	K = 0.11	21.8 ± 4.76	K = 0.54
3	7	28	45.57 ± 9.41	*p* = 0.328	21.43 ± 9.85	*p* = 0.943	24.28 ± 6.68	*p* = 0.762
>3	11	44	50.09 ± 8.12		19.72 ± 7.58		22.63 ± 5.54	
**How often he/she uses bathroom per day (24 h)**								
6–10 times	12	48	45.33 ± 9.53	K = 1.37	21.25 ± 8	K = 0.20	23.08 ± 6.04	K = 1.87
10–15 times	5	20	49.8 ± 9.2	*p* = 0.503	21.4 ± 8.99	*p* = 0.901	26.2 ± 6.06	*p* = 0.391
>15 times	6	24	50.17 ± 6.11		19.5 ± 8.9		22.33 ± 4.46	
**Use of aids**								
No	6	24	42.67 ± 9.02	U = 34	18.33 ± 7.94	U = 45	21.33 ± 5.08	U = 44
Yes	19	76	49.05 ± 8.11	*p* = 0.152	21.21 ± 7.73	*p* = 0.465	23.79 ± 5.66	*p* = 0.423
**Use of diuretics**								
No	9	36	44.66 ± 8.09	U = 54	19 ± 6.7	U = 32.5	21.77 ± 4.68	U = 57
Yes	16	64	49.12 ± 8.72	*p* = 0.322	21.37 ± 8.31	*p* = 0.610	24 ± 5.94	*p* = 0.412
**Use of sedatives**								
No	19	76	47.68 ± 8.33	U = 55	20.47 ± 7.26	U = 53.5	23.1 ± 5.17	U = 55.5
Yes	6	24	47 ± 10.26	*p* = 0.920	20.67 ± 9.79	*p* = 0.849	23.5 ± 7.09	*p* = 0.952

U = Mann‒Whitney U test; K = Kruskal‒Wallis test; * *p* < 0.05.

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
