# Peer review of "The Impact of Urinary Incontinence on Quality of Life: A Cross-Sectional Study in the Metropolitan City of Naples"

_geriatrics, 2020, doi:10.3390/geriatrics5040096_

Round 1

Reviewer 1 Report

Dear authors, this is an interesting manuscript on impact of urinary incontinence in Naples people. 

Unlikely the manuscript has several bias as the inadequate number of patients and the union of male and female patients. To improve the quality of this manuscript some change have to be done with the aim to reduce the impact of bias in the manuscript.

To better screen the kind of urinary incontinence I suggest to cite and introduce in the discussion this manuscript that analyze the costs and the incidence of pure stress urinary incontinence.

- Rubilotta et al, Pure stress urinary incontinence: analysis of prevalence, estimation of costs, and financial impact BMC Urol 2019 Jun 4;19(1):44. doi: 10.1186/s12894-019-0468-2.

Moreover, in the discussion should be cited also the counseling in patients with urinary incontinence as a pivotal part of decision making process between the treatment of these patients. I suggest to cite and read this manuscript:

- Counseling in urogynecology: A difficult task, or simply good surgeon-patient communication? Int Urogynecology J2018 Jul;29(7):943-948. doi: 10.1007/s00192-018-3673-8. Epub 2018 May 29.   This manuscript still has bias but with an accurate use of these two citations in the discussion it is possible to ameliorate the limits of the manuscript.

Author Response

Dear reviewer, thank you very much for your support and suggestions. Here's the changes we made:

Point 1: "To better screen the kind of urinary incontinence I suggest to cite and introduce in the discussion this manuscript that analyze the costs and the incidence of pure stress urinary incontinence"

Comment: we have wrote some sentences in lines 266-269 that add some relevant information regarding the reference suggested by you, which has been added to the reference list too.

Point 2: "Moreover, in the discussion should be cited also the counseling in patients with urinary incontinence as a pivotal part of decision making process between the treatment of these patients. I suggest to cite and read this manuscript"

Comment: we have wrote some sentences in lines 305-309 that add some relevant information regarding the reference suggested by you, which has been added to the reference list too. 

Thank you again very much for your support.

Please see details in the revised manuscript.

Benedetto Giardulli

Reviewer 2 Report

The Authors  adequately addressed  all previous criticisms.

Author Response

Dear reviewer,

Thank you again for your support and for your suggestions.

Have a nice day

Reviewer 3 Report

I want to thank you again for the opportunity to write a review. I found a few more grammatical errors that I corrected and suggested changes.

Line 37: this is the first time you mention the term „quality of life“ in the text so you should write the abbreviation in parentheses.

....and therefore it may decrease patients’ quality of life (QOL).

Line 75: I don't understand what the term „severity“ refers to, because „severity of patients“ is a bit awkward. Did you mean „Severity of illness/symptom“ or did you mean „The severity of the situation in which the patients find themself when they have UI symptoms“

If you meant „severity of UI symptoms“ please change the sentence to: To assess QOL and severity of UI symptoms in patients...

Line 82-83: the sentence is not grammatically correct so it should be written like this: ...it did not allow a holistic assessment of the patient considering his physical, emotional, social, economic, and spiritual needs.

Line 89-91: the sentence is awkward. This is my suggestion: The second part of the questionnaire was directed towards the clinicians with the intention to indicate the type of UI based on the information reported in the previous section and quantify the severity of the condition. This section was not intended to be filled in by patients.

In British English, generally ‘fill in a form’ is used, the verb ‘fill out’ meaning ‘to expand’, in the sense of putting on weight, but rarely in the sense of completing a form.

Line 96-97: the sentence is awkward. This is my suggestion: ...if the patients used orthopaedic or walking aids that supported them in walking, they were considered nonautonomous. I think that you don't need to specify the patients' sex (e.g. „him or her“ and „he or she“).

Line 102: Losses during the day:

I think it is necessary to specify that the term „losses“ refers to „episodes of urinary incontinence“. This term is most commonly used in scientific literature. Therefore please write „Episodes of UI during the day“

Line 104: Frequency of the episodes of UI

Line 144: 28 patients with a confirmed diagnosis of UI

Line 145: ...because they did not fill in the questionnaire

Line 153: ... and to fill in the questionnaire

Line 163: . Since the study presented was a brief report

Table 1. Please change „Stress“ to „Stress-„ and „Urge“ to „Urgency-“

Figure 2. Please change „Cronic bladder cystitis“ to „Chronic bladder cystitis“

Table 2.

For how much time has he/she been suffering from UI symptoms

The catheter is not absorbent aid so it would be more appropriate to use „UI aids used:“ instead of „Absorbent aids used:“

Line 183: Ethical approval was not required but anyway

Table 4. and Table 6. Please change „Urge“ to „Urgency-“

Line 222: ... and the severity of UI symptoms

Line 223-224: UI is the cause of embarrassment, especially for women

Line 228-230: Indeed, in the study, there was a considerable impairment of the Being domain in those whose highest educational level completed was a primary school.

Line 232: ...which included work

Line 232-234: This could be attributed to the fact that most of the women were unemployed or housewives, therefore, UI did not have a negative impact on the job.

Line 236: the sentence is awkward. This is my suggestion: BMI is strongly correlated with the severity of UI symptoms, and consequently with QOL because the increase of intra-abdominal pressure causes weakening of pelvic floor muscles and fascia.

Line 238: ...higher BMI is associated with more severe UI symptoms.

Line 244-246: the sentence is awkward. This is my suggestion: Therefore, this suggests that diuretic drugs cause more severe UI symptoms and consequent greater impairment of QOL.

Line 246-248: the sentence is awkward. This is my suggestion: This relationship can be explained by the increase of the volume of urine produced, and therefore by the increase in both the number of times the patient goes to the bathroom and the amount of involuntary urine leakage.

Line 265-266: please leave out „and“ and leave „or to the lack of correlation“

Line 266-269: the sentence is awkward. This is my suggestion:

If we add to this the unnecessary surgical procedures that many patients experience which could be prevented urodynamic investigation, the direct costs reimbursed by the Italian National Health System are even higher [30].

Author Response

Dear reviewer,

Thank you for your help and support. We have analysed all your comments and suggestions. As you can see from the revised version of the manuscript, we have applied all your suggestions, which we found very helpfull indeed. I hope to work again with you. 

Thank you again and have a nice day.

Dr. Benedetto Giardulli 

Round 2

Reviewer 1 Report

The suggested changes have been done improving the manuscript, and reducing the limits. 

Author Response

Dear reviewer, thank you again for your support and suggestions.

We hope to work again with you.

Have a nice day

- Dr. Benedetto Giardulli